**RESEARCH REPORT**

# Epidermal cell fusion promotes the transition from an embryonic to a larval transcriptome in *C. elegans*

**Owen H. Funk, Daniel L. Levy\* and David S. Fay\***

## ABSTRACT

Cell fusion is a fundamental process in the development of many multicellular organisms, but its precise role in gene regulation and differentiation remains largely unknown. The *Caenorhabditis elegans* epidermis, which comprises multiple syncytial cells in the adult, represents a powerful model for studying cell fusion in the context of animal development. The largest of these epidermal syncytia, hyp7, integrates 139 individual nuclei through processive cell fusion mediated by the fusogenic protein EFF-1. To explore the role of cell fusion in developmental progression and associated gene expression changes, we conducted transcriptomic analyses of *eff-1* fusion-defective *C. elegans* mutants. Our RNA-seq findings showed widespread transcriptomic changes including the enrichment of epidermal genes and molecular pathways involved in epidermal function during development. Single-molecule fluorescence *in situ* hybridization further validated the observed altered expression of mRNA transcripts. Moreover, bioinformatic analysis suggests that fusion may play a key role in promoting developmental progression within the epidermis. Our results underscore the significance of cell–cell fusion in shaping transcriptional programs during development.

KEY WORDS: Cell fusion, EFF-1, *C. elegans*, Epidermis, Syncytia, Transcription

## INTRODUCTION

Multinucleate syncytial cells are observed across a variety of species and biological contexts including in healthy tissues and in disease states such as cancer (Abmayr and Pavlath, 2012; Brooks et al., 2019; Calvert et al., 2016; Larsson et al., 2008; Ogle et al., 2005; Roper et al., 2011). The functional necessity for a syncytium varies among organisms and tissues; multiple nuclei may be required to metabolically support larger cells or to facilitate the rapid production of large amounts of mRNA for developmental processes or in response to extracellular cues (Cardamone et al., 2008; Deshpande et al., 2022; Dowling et al., 2021; Iosilevskii and Podbilewicz, 2021; Orr-Weaver, 2015). Multinucleation can arise through several paths, including cell–cell fusion of mononucleate cells (Aguilar et al., 2013; Alper and Podbilewicz, 2008; Chen et al., 2007; Demin et al., 2022; Ogle et al., 2005). This cellular fusion necessarily merges the cytoplasm of two distinct cells, but how the mixing of cytoplasmic factors changes the

regulatory environment of individual nuclei and thus affects the transcriptome within a newly formed syncytium is poorly understood (Kim et al., 2020; Liu et al., 2009; Petrany et al., 2020; Williams et al., 2022).

The *Caenorhabditis elegans* epidermis (or hypodermis) is an excellent model to address questions related to syncytial nuclei function. *C. elegans* has a well-characterized and largely invariant cell lineage in which over a third of somatic nuclei ultimately reside within syncytial cells (Alper and Podbilewicz, 2008; Podbilewicz and White, 1994). The adult worm epidermis comprises multiple syncytial cells, the largest of which is hyp7, which surrounds most of the adult body (Alper and Podbilewicz, 2008). hyp7 forms through progressive rounds of cell fusion, beginning with 23 embryonic fusion events that create the initial larval hyp7 before hatching. This process is completely dependent on cell–cell fusion mediated by the fusogen EFF-1 and is followed by the addition of 116 additional nuclei through EFF-1–mediated fusion events during larval development (Mohler et al., 2002; Podbilewicz et al., 2006; Shemer et al., 2004).

*eff-1* was discovered through genetic screens as the lone factor that is both necessary and sufficient to induce cell fusion in the *C. elegans* epidermis (Mohler et al., 2002; Shemer et al., 2004). Mutations in *eff-1* disrupt embryonic fusion events in the developing epidermal syncytium, leaving the epidermal progenitors unfused and embryonic cell boundaries intact. *eff-1* mutant worms exhibit variable morphological and developmental defects, highlighting the importance of cell fusion for proper body morphogenesis (Brabin et al., 2011; Koneru et al., 2021; Mohler et al., 2002; Podbilewicz et al., 2006).

Here, we explore the role of embryonic cell fusion in gene expression and differentiation in the developing epidermal syncytium through RNA-seq of worms containing reduction-of-function mutations in EFF-1, which block most epidermal fusion events in the embryo. Reduced fusion led to an increase in embryo-associated gene transcripts concurrent with a decreased expression of larval-associated transcripts in genes expressed specifically in the epidermis. These findings suggest a developmental delay in the acquisition of epidermal cell fates and a previously unreported role for cell fusion in activating gene regulatory programs associated with epidermal differentiation, adding to the growing list of functions for cell fusion and syncytialization.

## RESULTS AND DISCUSSION

### EFF-1–mediated cell fusion is essential for normal developmental progression and morphology in *C. elegans*

The discovery of EFF-1 as the primary fusogen governing most epidermal cell fusions in *C. elegans* originated from classical mutagenesis screens (Mohler et al., 2002; Podbilewicz et al., 2006; Shemer et al., 2004). This included two mutations – P183L (*hy21*) and S441L (*oj55*) – with variably penetrant fusion defects and temperature sensitivity. Both mutations appear to be partial reduction-of-function mutations rather than complete knockouts and are expected to contain

Department of Molecular Biology, University of Wyoming, Laramie, WY 82071, USA.

*Authors for correspondence (davidfay@uwyo.edu; dlevy1@uwyo.edu)

D.L.L., 0000-0002-7853-3275; D.S.F., 0000-0002-7599-4017

background mutations as a result of the mutagenesis (Mohler et al., 2002). Additional historical strains exist, including a likely null resulting from a large deletion (*ok1021*); however, this strain is also expected to contain background mutations caused by the mutagen-based deletion process (*C. elegans* Deletion Mutant Consortium, 2012). Moreover, the parental wild-type (WT; N2) strain used to derive all three of these *eff-1* mutants may differ slightly from our lab's N2 strain due to genetic drift. To overcome these limitations, we generated strains containing a putative null mutation in *eff-1* using CRISPR to place two stop codons in exon 1, which is present in all *eff-1* isoforms (P37Stop; Fig. 1A). This precise modification provides a clean, fusion-null baseline for comparison with our N2 control, potentially leading to a stronger reduction of function than the historical alleles.

Consistent with Mohler et al. (2002), *eff-1*(P37Stop) mutants displayed characteristic morphological defects, including a bulging anterior region, tail spike abnormalities and a dumpy phenotype that persisted into adulthood (Fig. 1B). To visualize fusion defects in *eff-1* mutants, we imaged apical junctions using AJM-1::GFP and confirmed that *eff-1* mutants had an unfused epidermis that persisted into adulthood (Fig. 1C-E). We note that the original characterization of the *eff-1* mutants reported a complete lack of fusion during late embryonic stages but partial fusion during larval development at temperatures below 25°C (Mohler et al., 2002). Unfused cells were still apparent in adult P37Stop and P183L worms, whereas the hyp7 syncytia were mostly fused in S441L adults (Fig. S1). The presence of morphological defects in S441L

adults suggests that consequences of early fusion defects may persist even if fusion is largely recovered at later stages.

We next characterized the developmental timing of *eff-1*(P37Stop) and the historical mutants with respect to the lengths of (1) embryonic development (time to hatch), (2) the first larval stage (L1), and (3) the transition from early L1 to reproductive adulthood. Although fusion events in the growing epidermal syncytia are initiated during late embryonic phases, the time to hatch was not affected in *eff-1* mutant strains (Fig. 1F). Conversely, all three *eff-1* mutants exhibited developmental delays in the lengths of L1 and the time to reach adulthood, although the delays in *eff-1*(P183L) and *eff-1*(P37Stop) were more pronounced than in *eff-1*(S441L) (Fig. 1F). Additionally, all three *eff-1* mutants were less synchronous than WT, and we observed a low frequency of larval arrest in the *eff-1*(P183L) and *eff-1*(P37Stop) mutants. Thus, although disruption of cell fusion does not alter the length of embryogenesis, it significantly alters the pace and synchrony of larval development.

## Disruption of cell fusion in embryonic development leads to consistent, widespread transcriptomic changes in *eff-1* mutants

The transition from embryo to larva is accompanied by widespread transcriptional rewiring as the newly hatched L1s begin locomotion and feeding and many tissues and cell types acquire their terminal post-mitotic fate (Boeck et al., 2016; Cao et al., 2017; Murray et al., 2008; Packer et al., 2019). To examine the role of cell fusion in the embryonic-to-larval transition, we performed RNA-seq on WT and

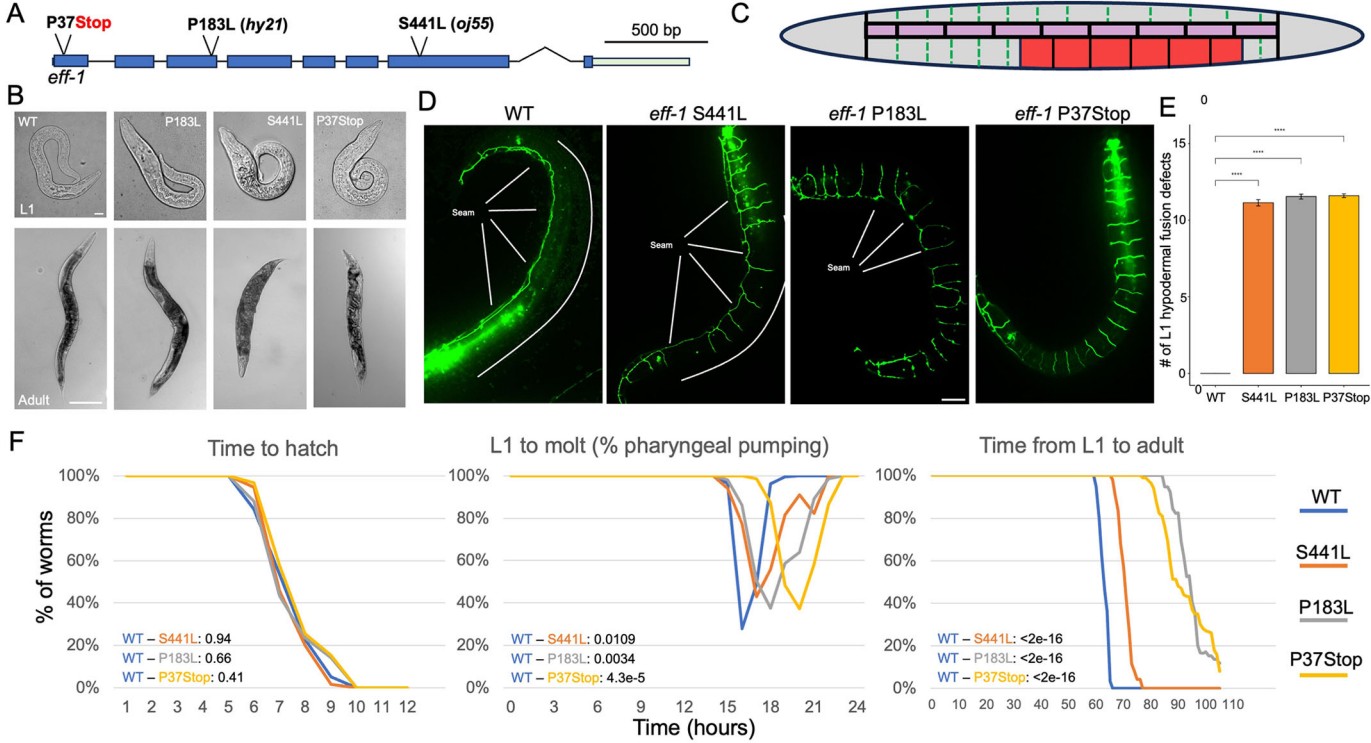

**Fig. 1. *eff-1* mutant worms display morphological defects and developmental delay.** (A) *eff-1* locus with CRISPR edit P37Stop and mutant alleles P183L (*hy21*) and S441L (*oj55*). (B) Brightfield imaging of L1 and adult N2 and *eff-1* mutants. Scale bars: L1, 10 µm; adult, 100 µm. (C) Schematic of L1 epidermis showing EFF-1–mediated fusion events (dashed green lines), seam cells (pink) and P cells (red). (D) Representative images of AJM-1::GFP fluorescence (apical cell–cell junctions in the epidermis) in N2 (WT) and *eff-1* mutant L1 larvae; lateral seam cells are labeled. Scale bar: 10 µm. (E) The average number of hyp7 fusion defects per genotype present at L1 (*t*-tests). ****P<0.0001 (Mann-Whitney U test, *n*≥40 per condition). (F) Developmental timepoints in N2 and *eff-1* mutants. Time to first molt was measured via cessation of pharyngeal pumping. Time to hatch and adulthood was measured by locomotion and sexual maturity, respectively. n of >40 per condition and timepoint. *P*-values for 1F (shown on the graphs) obtained by pairwise log-rank test between WT and each *eff-1* allele.

*eff-1* mutant L1s, which were synchronized by bleaching gravid adults and allowing released embryos to arrest via starvation-induced L1 diapause. For *eff-1*(P37Stop), differential expression analysis revealed 2046 upregulated and 912 downregulated differentially expressed genes (DEGs) relative to WT, with a false-discovery rate (FDR) cutoff of <0.05 and $\log_2$(fold change) cutoff of >0.5 or <–0.5 (Fig. 2A, Table S1). Additionally, we found 3534 upregulated and 2558 downregulated DEGs in *eff-1*(S441L) mutants and 6499 upregulated and 6829 downregulated DEGs in *eff-1*(P183L) mutants (Fig. S2A, Table S1). The larger number of DEGs observed in the classically isolated mutant strains is not entirely unexpected, as these strains were generated by random mutagenesis and from parental strains that may differ somewhat from our N2 strain. As such, some of the transcriptomic changes in these strains are likely attributable to background variations.

Despite these potential background differences, we observed a strong overlap between *eff-1*(P37Stop), *eff-1*(S441L), and *eff-1*(P183L), with 958 shared DEGs that were differentially regulated in the same direction (293 downregulated and 665 upregulated) (Fig. 2B,C). Hypergeometric analysis confirmed the significance of the overlaps (downregulated overlap *P*-value: 5.0e-53; upregulated overlap *P*-value: 8.6e-164), and the concordance remained robust even when more stringent cutoffs were applied (Fig. S3). Additionally, based on our analysis we would have expected only 253 triple-overlapping genes (189 upregulated and 64 downregulated genes) versus the 958 observed. Furthermore, $\log_2$-transformed fold-change data for the top DEGs from all three mutant strains also showed high concordance, with top genes shifting in the same direction regardless of the *eff-1* mutation (Fig. 2D). Euclidian distances between samples also confirmed transcriptome-wide similarities between *eff-1* point mutations and P37Stop as compared with WT (Fig. S2B). Notably, the P183L transcriptome appears to be particularly divergent, which may be due to a higher burden of background mutations, highlighting the utility of a clean engineered knockout.

Thus these data indicate a shared core transcriptomic response to the absence of epidermal cell fusion in L1 larvae, conserved across *eff-1* alleles and genetic backgrounds. The overlap in both the

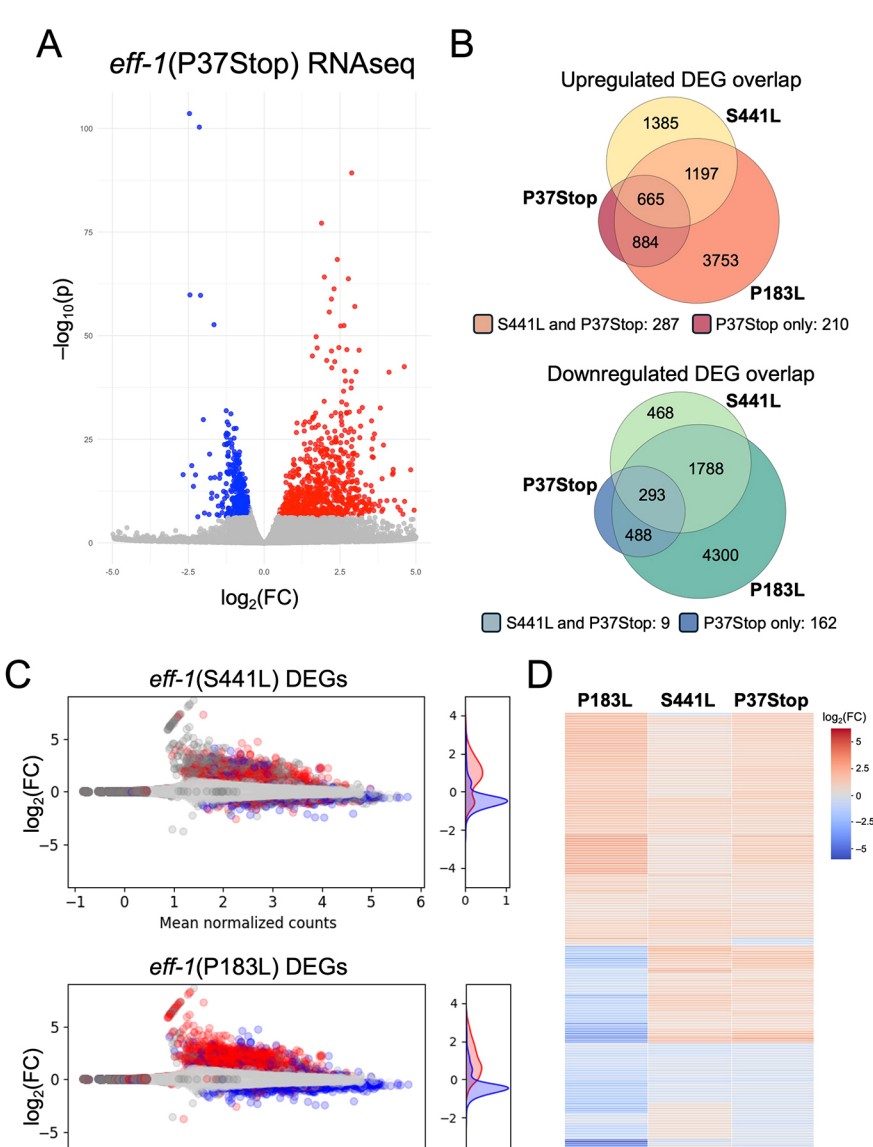

Fig. 2. RNA-seq of multiple *eff-1* mutants indicates common transcriptome response to fusion disruption. (A) Volcano plot of top DEGs (downregulated, blue; upregulated, red) for *eff-1*(P37Stop). (B) Overlap of significant DEGs [adjusted *P*-value<0.05, $\log_2$(FC)<−0.5 or>0.5] from P37Stop, S441L and P183L. (C) MA (minus average; Bland-Altman) plot of *eff-1*(P37Stop) mutant DEGs (relative to WT) with upregulated (red) and downregulated (blue) DEGs from S441L and P183L overlayed with corresponding genes. Right-side graphs: density of colored (shared DEGs) points. (D) Heatmap of top 200 DEGs shared between the *eff-1* mutants. FC, fold change.

identity of DEGs and the directionality of their fold changes across all *eff-1* mutants suggests a conserved transcriptional signature linked directly to the disruption of epidermal syncytialization. This highlights a fundamental set of gene expression changes that is consistently triggered by the failure of cells to fuse in the developing epidermis. Determining whether the affected gene sets are delayed in their onset or are permanently disrupted will require additional transcriptomic experiments along with careful staging to assess how fusion defects impact later larval development.

### Disruption of *eff-1*–mediated fusion alters epidermal cell expression

The hyp7 syncytium has many functions including the production of a new cuticle at each molting cycle, which occurs via the synthesis and secretion of apical extracellular matrix (aECM) proteins that include many collagens (Chisholm and Hsiao, 2012; Hendriks et al., 2014; Johnstone, 2000; Meeuse et al., 2020; Page and Johnstone, 2007). Consistent with a disruption of these functions, Gene Ontology (GO) analysis revealed a substantial enrichment for epidermal-specific processes, particularly extracellular proteins involved in molting and cuticle formation (e.g. structural constituent of cuticle, collagen-containing ECM) (Fig. 3A). This supports the notion that normal epidermal differentiation and function are disrupted or significantly delayed in *eff-1* mutants. Combined with the tissue enrichment analysis (Fig. 3C), these data suggest that the changes in gene expression observed in fusion-defective worms occur primarily among genes and pathways associated with the epidermis and epidermal functions relative to other cell and tissue types. Indeed, 89.6% of all shared DEGs have detectable expression in L1 epidermis, strengthening the conclusion that the hypodermal lineage is the primary site of transcriptional impact.

In addition to the curated set of GO terms on WormBase, we observed an enrichment in embryonic precuticle- or 'sheath'-associated genes that accompanied the decrease in L1-specific genes (Fig. 3B). The sheath is the first precuticle formed during embryonic development before secretion of the first cuticle (Costa et al., 1997; Mancuso et al., 2012; Priess and Hirsh, 1986; Sundaram and Pujol, 2024). Precuticle gene expression at the L1 stage in *eff-1* mutants suggests a delay in the transition from precuticle to cuticle gene expression programs. This finding, coupled with the observed downregulation of L1-specific collagen genes, points toward an overall impairment or delay in the transcriptional wiring necessary for proper L1 epidermal maturation and function.

To validate and visualize changes in the expression of specific DEGs, we used single-molecule inexpensive fluorescence *in situ* hybridization (smiFISH) of two DEGs identified in our RNA-seq data. *col-42* encodes a structural collagen largely restricted to larvae and adults and was significantly downregulated in both *eff-1*(S441L) and *eff-1*(P183L) mutants (Fig. 3D). *col-42* is expressed during early larval development, but smiFISH imaging showed almost no detectable signal in *eff-1* mutants at L1 (Fig. 3D). Conversely, *rhy-1* is a transmembrane acyltransferase that is expressed in the embryonic epidermis (Ma et al., 2012; Shen et al., 2006). Consistent with the transcriptomic data, the number of *rhy-1* mRNA fluorescent puncta was elevated in *eff-1* mutants (Fig. 3D).

*eff-1* phenotypes, such as the developmental delay, could be due to a combination of autonomous and non-autonomous effects including defects in other tissues. We thus looked for overlap between gold-standard lists of tissue-specific genes from small-scale single-tissue experiments and the *eff-1* DEGs (Kaletsky et al., 2018). We found a significant enrichment among downregulated DEGs for transcripts associated with the hyp7 syncytium, epidermis

and epithelial system, over non-syncytial tissues (Fig. 3C), consistent with a model in which unfused hyp7 progenitors fail to upregulate genes normally expressed in the L1 syncytial epidermis. This suggests that the observed phenotypic and transcriptomic changes are, in large part, the direct consequence of an unfused epidermis rather than of a systemic developmental delay. Nevertheless, our data indicate the presence of some non-autonomous effects, such as those observed in body wall muscle cells adjacent to epidermal cells. Collectively, our data support the central finding that fusion plays a crucial role in promoting the timely differentiation and function of the epidermis based on hallmark changes in gene expression.

### *eff-1* mutant transcriptional changes are consistent with a developmental delay

In addition to shared pathways and functions associated with the DEGs across *eff-1* mutations, we found a striking enrichment in DEGs associated with the transition from embryonic to larval development. By examining the developmental trajectory of DEGs at timepoints throughout the WT worm lifecycle from modENCODE (Gerstein et al., 2010; Hutter and Suh, 2016), we found an enrichment in L1-associated genes in the downregulated DEG set, along with upregulated DEGs showing substantial enrichment for genes normally expressed in the late embryo (Fig. 4A,D). Furthermore, genes that were not differentially expressed (FDR>0.05) in the *eff-1* mutants were predominantly those for which expression levels remain relatively stable across normal development. These observations strongly suggest that the normal transcriptional shift from late-embryonic to L1 larval epidermal expression programs is substantially delayed in worms deficient for EFF-1–mediated fusion.

To explore the typical trajectory of the affected genes, we then used a linear regression of WT gene expression data across multiple embryonic and larval timepoints to define gene sets that are developmentally upregulated or downregulated during the transition from late-embryonic to larval stages. Comparing the average fold changes of these developmentally dynamic gene sets in *eff-1*(P37Stop) mutants revealed a net increase in the expression of developmentally downregulated (i.e. typically embryonically enriched) genes and a corresponding decrease in the expression of developmentally upregulated (i.e. typically larvally enriched) genes, irrespective of whether individual genes met the strict criteria for differential expression (Fig. 4B). This broader analysis reinforces the conclusion that the absence of cell fusion leads to a significant delay in the acquisition of the appropriate L1 epidermal transcriptome.

The transition from an embryonic to larval transcriptome is facilitated by many transcription factors, including ELT-3, an embryonically enriched GATA transcription factor with roles in transcriptional activation of epidermis-specific genes and developmental pathways (Broitman-Maduro et al., 2022; Gilleard and McGhee, 2001; Gilleard et al., 1999). An *elt-3* construct that is expressed in only a subset of epidermal precursor cells is used as a reporter for fusion, or lack thereof, in *eff-1* mutant worms (Shinn-Thomas et al., 2016). Intriguingly, a large swath of ELT-3–regulated genes identified through the TF2DNA database were significantly downregulated in *eff-1* mutant worms (Fig. 4C). Examination of the ELT-3 chromatin immunoprecipitation sequencing (ChIP-seq) signal across all upregulated, downregulated and unchanged genes also showed an enrichment of downregulated genes. Of the downregulated DEGs, those associated with ELT-3 exhibited more negative fold changes and more significant *P*-values, suggesting that ELT-3–regulated genes might be particularly affected by the loss of fusion

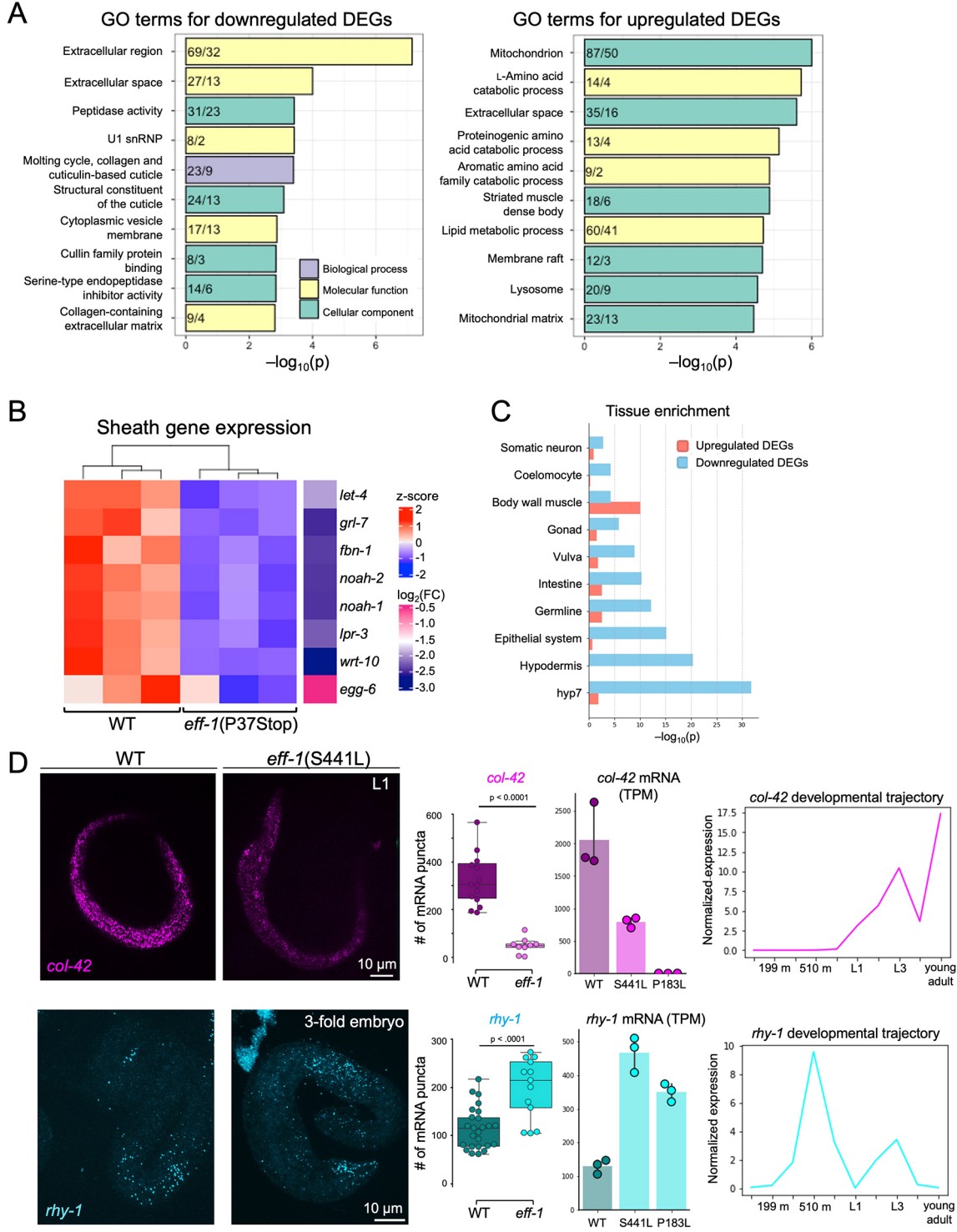

**Fig. 3. The null *eff-1* mutation affects diverse functions.** (A) GO term enrichment with DEGs from *eff-1*(P37Stop) RNA-seq. Colors indicate GO term category (see key). Numbers associated with each bar indicate observed/expected genes for each GO term. (B) Heatmap of *z*-score–transformed mean expression and log₂(FC) values for selected sheath genes in WT and *eff*-1(P37Stop) L1s. (C) Tissue enrichment analysis of DEGs based on gold-standard small-scale enrichment experiments. DEGs were intersected with lists of tissue-specific genes; *P*-values calculated based on hypergeometric significance. (D) Representative images of *col-42* and *rhy-1* smiFISH hybridization in WT and e*ff*-1(S441L) L1s and 3-fold-stage embryos, respectively. mRNAs were quantified based on the number of fluorescent puncta (left graphs). RNA-seq counts (transcripts per million; TPM) for WT and *eff-1* mutants for *col-42* mRNA in L1 larvae and *rhy-1* mRNA in embryos (middle graphs). Developmental trajectory of *col-42* and *rhy-1* expression in WT animals taken from modENCODE expression data (Gerstein et al., 2010). Data are presented as box plots overlaid by the individual data points. The box represents the interquartile range (25th to 75th percentiles), with the center line denoting the median. Whiskers extend to the minimum and maximum values within 1.5 times the interquartile range. Statistical significance was determined using an unpaired two-tailed Student's *t*-test. Error bars (middle graphs) show s.e.m.; ≥9 animals/condition.

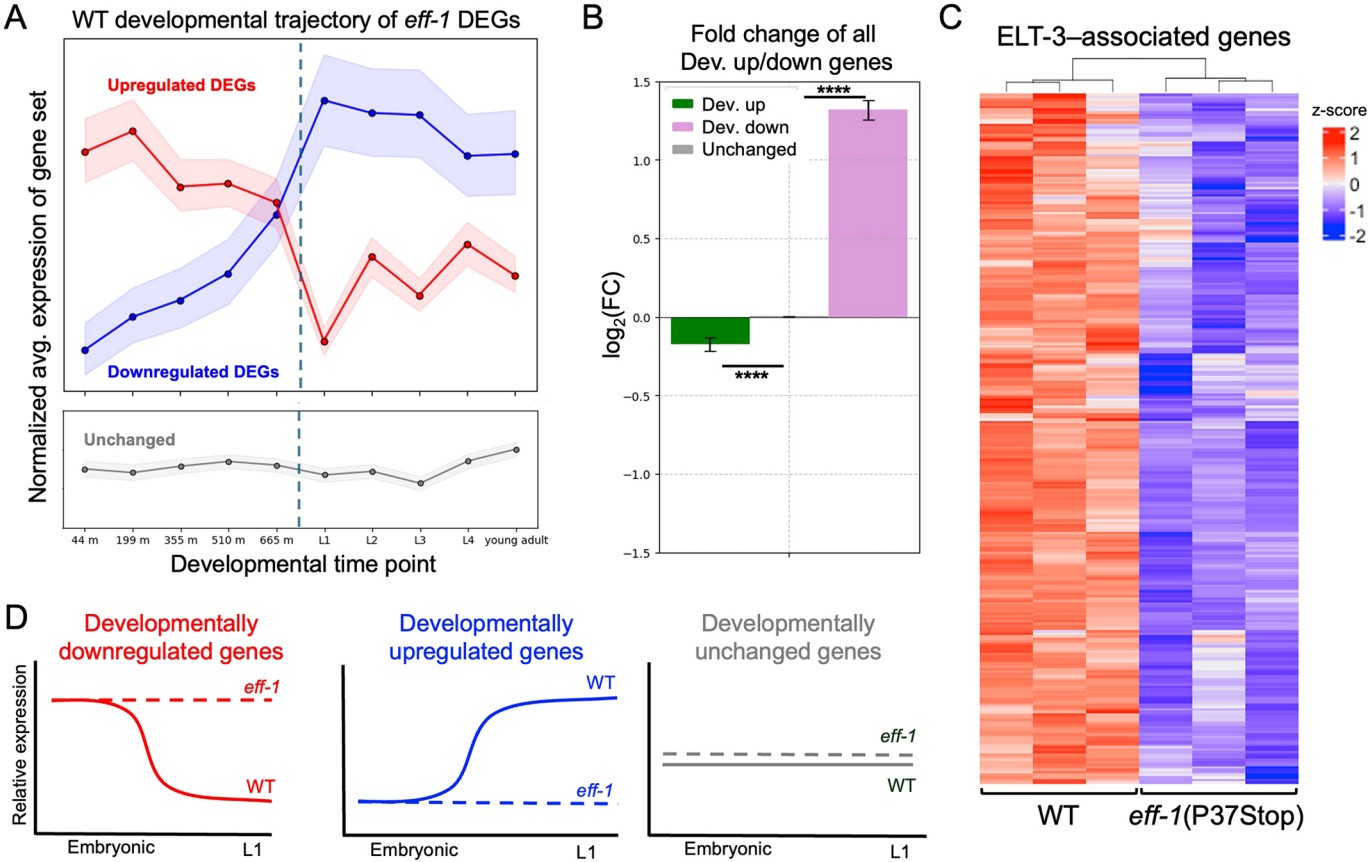

**Fig. 4. Fusion mutants fail to acquire the L1 epidermal transcriptome.** (A) Normal (WT) developmental trajectories of upregulated (red) or downregulated (blue) genes in *eff-1* mutants. Shaded region, 95% confidence interval. Dashed line, approximate hatching time of 800 min. (B) Genes were binned using a linear regression of modENCODE data to determine transcripts that went up (green), down (purple) or stayed consistent (gray) across WT development. Average $\log_2$-transformed fold change RNA-seq data from *eff-1*(P37Stop) for each category of genes are shown. Error bars show s.e.m.; Mann–Whitney *U*-test, ****$P$<0.00001. (C) Heatmap of *z*-score–normalized DEG expression for genes identified as ELT-3-regulated from the TF2DNA dataset (Pujato et al., 2014). (D) Schematic of WT and *eff-1* mutant gene expression profiles across development.

(Fig. S4). This downregulation of ELT-3 target genes in *eff-1* mutants provides a potential mechanistic link between the failure of cell fusion and the observed developmental delay. Cell fusion may be necessary for appropriate stage-specific activity of key developmental regulators such as ELT-3, which may be expressed in only a subset of cells contributing to epidermal syncytia.

Our work underscores the crucial role of EFF-1–mediated cell fusion in the developmental progression of *C. elegans* and highlights its importance for timely activation of the L1 larval epidermal transcriptome. Our findings suggest that the process of fusion itself may be a key mechanistic trigger promoting the shift from the embryonic to the L1 transcriptional landscape in fusion-driven syncytia such as hyp7. Although these results show the importance of syncytial formation for proper expression of developmental genes, precisely how fusion orchestrates transcriptional maturation remains an open question. Future investigations will be needed to unravel the molecular mechanisms involved and to identify specific factors that drive the identity and transcriptional programs of developing syncytial nuclei.

## MATERIALS AND METHODS
### Strains used and maintenance
*C. elegans* strains were maintained according to standard protocols and were propagated at 22°C. The two *eff-1* mutant strains used here were WH171 (*eff-1*(*oj55*); jcIs1 *[ajm-1::GFP+unc-29(+)+rol-6(su1006)]*) and BP76

(*eff-1*(*hy21*); jcIs1 *[ajm-1::GFP+unc-29(+)+rol-6(su1006)]*). MH1384 (*[kuIs46 (ajm-1::GFP; unc-119(+))]*) was used to compare WT AJM-1:: GFP expression to that in *eff-1* mutants. The P37Stop strain was generated using CRISPR via a previously described method (Ghanta et al., 2021). Briefly, RNPs loaded with gRNA (GGTGTCTTGGAACAGTGTGG) were injected along with repair template designed to add two stop codons along with an SpeI recognition site into the gonads of animals expressing *kuIs46*[AJM-1::GFP] and *bqSi640*[*dpy-7p*::FRT::mCherry::*his-58*::FRT:: GFP::*his-58*].

### Developmental timing analysis
Time to hatch was determined by plating young adult worms, allowing them to lay eggs for 2 h, and then removing the adults and measuring the percentage of unhatched eggs that remained over time. Time to molt was determined by plating synchronized L1 larvae and watching for cessation of pharyngeal pumping to mark the beginning of the L1-L2 molt. Time to adulthood was measured by plating synchronized L1 larvae and noting the presence of embryos in the gonads. All experiments were carried out at 22°C. A log-rank test was performed for each developmental metric, and time to 50% cessation of pumping was used for the L1 to first molt analysis.

### Microscopy
Confocal images for Fig. 1 were taken using CellSens on an Olympus Spin-SR, Spinning Disc, Super Resolution System using an inverted Olympus IX83 microscope and Yokogawa W1 spinning disc. smiFISH images were taken of worms fixed in methanol/acetone as described (Parker et al., 2021).

Larvae for Fig. 1 images were prepared in 10 mM levamisole on 3% agarose pads.

## RNA isolation and RNA-seq

L1s were synchronized by bleaching gravid adults and allowing embryos to arrest in M9 buffer for 24 h. These suspensions were further inspected to ensure no unhatched but viable embryos remained. L1 embryos were spun down and lysed for RNA collection via the Direct-zol kit from Zymo. RNA quality was confirmed using an Agilent TapeStation with minimum RNA integrity numbers (RINs) of 8.0. mRNA libraries were constructed using TruSeq Library Prep Kits from Illumina and were sequenced on an Illumina NextSeq P3 flow cell. For each RNA-seq experiment, three biological replicates were performed in parallel starting from individual 150-mm plates. Three replicates were performed for S441L, P183L and WT (N2) worms in parallel, followed by an additional experiment where three P37Stop and three WT replicates were performed.

## Transcriptomic analysis

Paired-end reads were processed as follows. Low-quality reads were filtered using fastp, and their quality was assessed using fastQC (0.12.1). Reads were aligned using STAR and pseudoaligned using Salmon to the *ce11* reference genome, duplicates were removed using picard, and counts were collected using htseq (Anders et al., 2015; Dobin et al., 2013; Patro et al., 2017). Differential expression was determined using DESeq2 (v.1.42.1), with counts filtered for >1 TPM across all samples. DEGs were defined as genes with a Benjamini–Hochberg adjusted $P<0.05$ and $\log_2$(fold change)>0.5 or<−0.5 (processed files in Table S1). Heatmaps were generated using the seaborn clustermap function using Euclidean distance to obtain hierarchal linkages and *z*-score normalization by rows. Intersectional analysis was carried out by collecting modENCODE developmental expression data (Gerstein et al., 2010; Hutter and Suh, 2016). Upregulated, downregulated and unchanged classifications were determined by performing a linear regression for expression data across developmental time points and collecting genes with a derived slope of >1 (developmentally upregulated) or <−1 (developmentally downregulated). Genes with a derived slope from 1 to −1 were categorized as unchanged. Raw and processed data files have been deposited in GEO under accession number GSE301850.

## GO term analysis

GO term enrichment was carried out using topGO (v.2.54.0) on ontologies from WormBase using the 'weight01' algorithm, and significance was assessed with a Fisher's statistic. Genes found to be differentially expressed (FDR<0.05) in both S441L and P183L were used as the test set, and the background gene set was specified as genes with a base mean TPM of >1 in all three transcriptomes.

## Tissue enrichment

Gold-standard tissue-specific gene sets were obtained from https://worm.princeton.edu/ (Kaletsky et al., 2018) and compared to DEGs from the *eff-1* mutant RNA-seq data. Fischer's exact test was used to calculate enrichment significance.

## smiFISH

smiFISH was carried out as previously described (Parker et al., 2021). Probe sets were designed against *col-42* and *rhy-1* using Oligostan with FLAP-X extensions (Tsanov et al., 2016). The FLAP-X secondary probes were labeled with Quasar 670 from Biosearch Technologies. Quantification was carried out in FIJI by manually drawing designated regions of interest around embryos and L1s of appropriate age and sufficient fixation, thresholding across experimental replicates, and using the 'analyze particles' option.

## ELT-3 ChIP-seq analysis

ELT-3–associated genes were obtained from the TF2DNA dataset (Pujato et al., 2014) and compared to our experimentally derived DEGs. L1 ELT-3 ChIP-seq data from modENCODE was downloaded as read-depth–normalized bigwig files. Heatmaps were generated with deeptools

(v.3.5.6) (Ramírez et al., 2016). Up- and downregulated gene sets were designated from the *eff-1* P37Stop dataset with the same criteria [adjusted *P*-value<0.05, $\log_2$(fold change)>0.5 or<−0.5]. Unchanged genes were taken as a random subset of 5000 genes with adjusted *P*>0.2.

## Acknowledgements

We thank Amy Fluet for editing this manuscript, the Nishimura lab for guidance and implementation of smiFISH, and the Sundaram and Murray labs for insightful conversation and feedback on the results. We also thank the Center for Advanced Scientific Instrumentation (CASI) core for assistance with confocal microscopy, and the University of Minnesota Genomics Center for processing and running samples for RNAseq.

## Competing interests

The authors declare no competing or financial interests.

## Author contributions

Conceptualization: D.S.F., O.H.F., D.L.L.; Data curation: O.H.F.; Formal analysis: D.S.F., O.H.F., D.L.L.; Funding acquisition: D.S.F., D.L.L.; Investigation: O.H.F.; Methodology: O.H.F.; Project administration: D.S.F., D.L.L.; Resources: D.S.F., D.L.L.; Supervision: D.S.F., D.L.L.; Validation: O.H.F.; Visualization: O.H.F.; Writing – original draft: O.H.F.; Writing – review & editing: D.S.F., O.H.F., D.L.L.

## Funding

This work was supported by the National Institutes of Health grants GM136236 and GM125091 (to D.S.F.) and GM134885 (to D.L.L.). This project was also supported by an Institutional Development Award (IDeA) from the National Institute of General Medical Sciences of the National Institutes of Health under P20GM103432. Open Access funding provided by the University of Wyoming. Deposited in PMC for immediate release.

## Data and resource availability

Raw and processed data files have been deposited in GEO under accession number GSE301850. All other relevant data and details of resources can be found within the article and its supplementary information.

## The people behind the papers

This article has an associated 'The people behind the papers' interview with some of the authors.

## Peer review history

The peer review history is available online at https://journals.biologists.com/dev/lookup/doi/10.1242/dev.205089.reviewer-comments.pdf

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
