## [Peer Review File · Development (Cambridge, England)]

Epidermal cell fusion promotes the transition from an embryonic to a larval transcriptome in *C. elegans*

Owen H. Funk, Daniel L. Levy and David S. Fay
DOI: 10.1242/dev.205089

Editor: Swathi Arur

Review timeline

Original submission:	7 July 2025
Editorial decision:	8 August 2025
First revision received:	8 October 2025
Accepted:	10 November 2025

Original submission

First decision letter

MS ID#: dev.205089

MS TITLE: Epidermal cell fusion promotes the transition from an embryonic to a larval transcriptome in *C. elegans*

AUTHORS: David S. Fay, Owen H. Funk and Daniel L. Levy

Dear Dr Fay,

I have now received all the referees reports on the above manuscript, and have reached a decision. The referees' comments are appended below, or you can access them online: please go to *********.

The overall evaluation is positive and we would like to publish a revised manuscript in Development, provided that the referees' comments can be satisfactorily addressed. Please attend to all of the reviewers' comments, all of which are textual, in your revised manuscript and detail them in your point-by-point response. If you do not agree with any of their criticisms or suggestions explain clearly why this is so. Please note that the revision will be reviewed by one or both the reviewers.

Reviewer 1

Advance summary and potential significance to field

This manuscript describes a potentially paradigm-shifting observation about cell fusion during development. The authors analyze three alleles of the fusogen gene *eff-1* in *C. elegans* and show that the failure of epidermal cells to fuse in these strains is associated with a delay in subsequent developmental events (molting and progression to adulthood), as well as a stage-inappropriate "early" gene expression profile in L1 larvae. While the paper doesn't really demonstrate mechanism, the concept is potentially paradigm shifting and novel.

The crux of the paper's impact is the idea that cell fusion promotes the temporal progression of epidermal differentiation. The data are all consistent with this idea, but other explanations exist, for example the dysregulated genes could be constitutively up/downregulated rather than

temporally delayed. For example, the smiFISH validation of *col-42* and *rhy-1* is nice but a time course of their expression could answer the question of whether the changes represent temporal differences vs general up/down regulation.

Other comments

According to WormBase there is a larger deletion allele available (*ok1021*) - given the effort described here to make a null allele, this allele is at minimum worth mentioning. Figure 1D would benefit from quantification to compare alleles and Figure 1E would benefit from statistical comparisons.

For Figure 2, the methods state these were starvation-arrested L1 larvae which should be mentioned in the Results for . Also, in the methods details like how long were embryos incubated in M9 prior to the RNA collection should be provided. Fig. S1B suggests three biological replicates per strain but this is not stated explicitly, and details are needed about whether these were grown in separate populations/days etc.

The number of DE genes identified by RNA-seq are quite large, especially for S144L for which 2/3 of all genes are DE. This is likely close to all genes expressed at this stage. This raises questions of whether some of the smaller effect size changes may be due to technical factors such as normalization artifacts or differences in tissue volume (see below) (in addition to genetic background which, as mentioned, is also a possible explanation). I would love to see an analysis in a supplemental figure of the number of regulated genes for each allele and overlap at a range of higher fold changes or minimum expression level thresholds. Another approach (overlapping somewhat with Fig. 3) would be to assess likely indirect effects such as normalization is to use existing single cell datasets to identify genes expressed exclusively in fusing vs non-fusing tissues; the latter could give a conservative estimate of the false positive rate (recognizing that non-autonomous or other explanations are also possible especially for the upregulated genes).

145: "strong overlap between *eff-1*(P37Stop), *eff-1*(S441L), and *eff-1*(P183L), with 958 shared DEGs" - please comment on how much this overlap is enriched compared to expectation given the number of genes.

I was confused by Fig S1B - the authors claim is that this figure supports the idea that point mutations more similar to each other than WT but it looks to me like WT is more similar to S441L and P37Stop and P183L more divergent (at least with this metric). Maybe this can be clarified or explained?

Perhaps related to the tissue volume comment above, the most strongly upregulated gene group in Fig. 3 is muscle. One explanation would be that *eff-1*(-) leads to decreased growth of epidermis, and a secondary consequence is in whole animal RNA-seq the proportion of transcripts from muscle is higher (since RNA-seq measurements are really proportions). Are there data to support this (or other explanations)?

The logic for the ELT-3 section seems a bit circular to me- ELT-3 targets are known to be enriched in hyp, hyp genes tend to be downregulated in *eff-1* mutants. Is there anything to suggest ELT-3 targets are more downregulated than other hyp-enriched genes? If so this would be useful to incorporate into Fig. S2.

Reviewer 2

Advance summary and potential significance to field

This manuscript approaches an interesting and poorly understood question regarding the role of cell fusion in gene regulation and differentiation. The authors take advantage of a large-scale, developmentally programmed set of cell fusions during *C. elegans* embryogenesis in which 23 embryonic cells fuse to create the initial larval hyp7 epidermal syncytium. They generated a predicted null allele in the fusogen gene, *eff-1*, and compared it to two classic alleles, which are likely partial reduction-of-function alleles. Their null allele displayed comparable larval

developmental delay to the two hypomorphs and the authors found a common set of misregulated genes in the *eff-1* mutants through RNA-seq. The cell fusion defective *eff-1* mutants have a gene expression pattern consistent with a developmental delay and failure to establish a larval transcriptome. Overall, this is an interesting and rigorous study and suitable for Development once the following issues are addressed.

Major points

1. Lines 103-104. The authors indicate that the cell fusion defects in their *eff-1* null mutant persist to adulthood, but only L1 images are shown. Given that the hypomorphs were reported to have partial cell fusion during larval development, it would be valuable to include data showing the difference between adult null and hypomorph cell fusion.
2. Given the developmental delay of the *eff-1* mutants and the large-scale oscillatory gene expression in the epidermis how did the authors ensure that the RNA samples were harvested at comparable developmental timepoints? This information is critical for interpreting the RNA-seq and smiFISH data.
3. There are a number of translational fusions available for pre-cuticle components and various cuticle components, many of which are dysregulated in *eff-1* mutants. It would substantially strengthen the paper to look at localization of pre-cuticle components such as *noah-1* and *noah-2*, embryonic cuticle components like *dpy-14* or *dpy-17* (should be in L1 cuticle but not expressed in L1), and markers for other structures like furrows and annuli. Those experiments would demonstrate how this embryonic-like gene expression program affects cuticle structure.

Minor points

1. In hindsight it would have been better to generate clean *eff-1* P183L and S441L mutations by CRISPR in an N2 background to remove the confounding variable of other background mutations. That work is beyond the scope of this paper as the work is rigorous, but it would have allowed insight into gene regulation differences produced by the null vs the hypomorphic alleles.
2. I may have missed it, but I didn't see any description/discussion of the Time to Hatch data in Fig 1E. It's worth noting that embryonic development seems to proceed normally, but larval development is delayed.
3. The species and gene names in the references need italicizing.

First revision

Author response to reviewers' comments

We thank the reviewers and the editor for their time and valuable input on our manuscript. We have now addressed the reviewer's comments (text in blue) including changes to the text and several figures. We note that additions to the text were necessarily kept to a minimum to avoid going over the word limit. We look forward to hearing back.

Reviewer 1: This manuscript describes a potentially paradigm-shifting observation about cell fusion during development. The authors analyze three alleles of the fusogen gene *eff-1* in *C. elegans* and show that the failure of epidermal cells to fuse in these strains is associated with a delay in subsequent developmental events (molting and progression to adulthood), as well as a stage-inappropriate "early" gene expression profile in L1 larvae. While the paper doesn't really demonstrate mechanism, the concept is potentially paradigm shifting and novel. The crux of the paper's impact is the idea that cell fusion promotes the temporal progression of epidermal differentiation. The data are all consistent with this idea, but other explanations exist, for example the dysregulated genes could be constitutively up/downregulated rather than temporally delayed. For example, the smiFISH validation of *col-42* and *rhy-1* is nice but a time course of their expression could answer the question of whether the changes represent temporal differences vs general up/down regulation.

We thank the reviewer for this insightful comment. There remains the possibility that these genes do not recover and are constitutively dysregulated throughout the life of the worm. Unfortunately, smiFISH signal deteriorates as the cuticle becomes thicker and more established across larval

development, a problem we've discussed at length with our collaborators in the Nishimura lab at Colorado State University where smiFISH in worms was pioneered. Furthermore, to definitively answer this question we would want to examine the entire transcriptome to see which transcripts exhibit prolonged disruption vs. eventual recovery. This would involve additional transcriptome sequencing, along with careful staging of mutants and wildtype to best match development stage. As such, these experiments are beyond the scope of the current study. Nevertheless, this is an important point for discussion, and we have now added additional text to this effect on lines 176-179 to address this possibility.

According to WormBase there is a larger deletion allele available (ok1021) - given the effort described here to make a null allele, this allele is at minimum worth mentioning. We now mention this strain in lines 98-101. We note that the consortium-based deletions were generated using mutagens, which would lead to numerous additional background mutations.

Figure 1D would benefit from quantification to compare alleles and Figure 1E would benefit from statistical comparisons.

We now provide quantification for Figure 1D (now panel 1E) and performed statistical analyses for time-to-event for the three developmental metrics used (now Figure 1F).

For Figure 2, the methods state these were starvation-arrested L1 larvae which should be mentioned in the Results.

We have added to the methods section detailing the procedures used and included this methodological point in the results section of the paper. (lines 141-143 and 326-328).

Also, in the methods details like how long were embryos incubated in M9 prior to the RNA collection should be provided.

We have amended the methods to include this information (lines 322-325).

Fig. S1B suggests three biological replicates per strain but this is not stated explicitly, and details are needed about whether these were grown in separate populations/days etc.

We have revised the methods to include this information in more detail.(lines 328-331)

The number of DE genes identified by RNA-seq are quite large, especially for S144L for which 2/3 of all genes are DE. This is likely close to all genes expressed at this stage. This raises questions of whether some of the smaller effect size changes may be due to technical factors such as normalization artifacts or differences in tissue volume (see below) (in addition to genetic background which, as mentioned, is also a possible explanation). I would love to see an analysis in a supplemental figure of the number of regulated genes for each allele and overlap at a range of higher fold changes or minimum expression level thresholds.

We thank the reviewer for this astute observation and now include these analyses in Fig. S3. There are indeed many DE genes that meet standard thresholds, and we have now conducted intersections with the much more stringent cutoffs using adjusted p-values of 0.05, 1e-7, 1e-13, and 1e-20 and log2 Fold change cutoffs of 0.5, 1.0, 1.5, and 2. This revealed significant overlap of all 3 alleles for all cutoffs in the down genes, and all but the most stringent cutoffs in up genes. This strengthens our argument that the transcriptomic changes and overlaps, including some of the smaller fold-changes, are likely to be the specific consequence of *eff-1* disruption. We now mention this in the text on lines 157-162.

Another approach (overlapping somewhat with Fig. 3) would be to assess likely indirect effects such as normalization is to use existing single cell datasets to identify genes expressed exclusively in fusing vs non-fusing tissues; the latter could give a conservative estimate of the false positive rate (recognizing that non-autonomous or other explanations are also possible especially for the upregulated genes).

We thank the reviewer for this thoughtful suggestion. While there are several excellent single-cell resources available for the worm, there are two primary impediments to using these data for comparisons in this study. 1) There are no L1 datasets to our knowledge, and hypodermis gene expression patterns change significantly throughout larval development into adulthood. 2) Through our exploration of cell fusion, we have found that the single-cell resources available largely lack data for syncytial cells, likely due to the large and unusual shape of these cells that

precludes flow cytometry used in single-cell workflows. As such, even if we were to use the single-cell data available, we would likely miss many important genes expressed in the hypodermal syncytia. For this reason, we elected to use bulk RNAseq data for comparisons based on tissue reporter strains with well characterized hypodermal markers. While no comparison is perfect, we felt this approach was the best way to address if epidermal genes were particularly affected. Having single-nuclei RNAseq data would be incredibly useful and would be the logical next step considering the current work, but a much larger undertaking. This is something we are planning for future studies.

145: "strong overlap between *eff-1*(P37Stop), *eff-1*(S441L), and *eff-1*(P183L), with 958 shared DEGs" - please comment on how much this overlap is enriched compared to expectation given the number of genes.

We thank the reviewer for this comment and have included hypergeometric analysis along with p values for these comparisons. Additionally, as mentioned above, we have performed hypergeometric tests for a variety of adjusted p-values and fold changes to assess the robustness of our claim of significant overlap, leading to Fig S3 as well as lines 160-162. "Additionally, based on our analysis we would have expected only 253 triple-overlapping genes (189 upregulated and 64 downregulated genes) versus the 958 observed."

I was confused by Fig S1B - the authors claim is that this figure supports the idea that point mutations more similar to each other than WT but it looks to me like WT is more similar to S441L and P37Stop and P183L more divergent (at least with this metric). Maybe this can be clarified or explained?

The P183L point mutant indeed appears more divergent from P37Stop and S441L than P37Stop and S441L do from each other. This could be due to several factors but given that these mutants were generated by random mutagenesis, it is likely that there are additional confounding mutations in the P183L background. This potential problem further highlights the utility of creating the stop allele in our own genetic background, to facilitate a more precise investigation of the specific transcriptomic effects of disrupting *EFF-1*. Because of this, we focus the rest of the paper on the analysis of genes affected in the P37Stop strain. We have now added language to this effect to the manuscript (lines 166-168).

Perhaps related to the tissue volume comment above, the most strongly upregulated gene group in Fig. 3 is muscle. One explanation would be that *eff-1*(-) leads to decreased growth of epidermis, and a secondary consequence is in whole animal RNA-seq the proportion of transcripts from muscle is higher (since RNA-seq measurements are really proportions). Are there data to support this (or other explanations)?

This comment raises several intriguing possibilities. As was rightly pointed out, RNAseq is inherently proportional. If we were simply seeing a decrease in epidermis tissue and associated RNAs, we would expect a modest increase in all other tissue types, rather than in individual tissue types as we observe. As the body wall muscle cells border the hypodermis, this does raise an interesting possibility that these cells might increase in size in unfused animals. We didn't observe obvious significant changes to the size of the epidermis in *eff-1* mutants, but we haven't looked closely at muscle cell volume. Alternatively, there might be some molecular confusion in the unfused cells, leading to aberrant transcription of muscle-associated genes, but this would require single cell/nuclei transcriptomics, additional smiFISH, and/or reporter lines to explore fully. Ultimately, we don't have a compelling answer for why some muscle genes become upregulated, and whether the transcriptomic source of these changes is affected muscle cells or unfused syncytial cells. We now mention that some non-autonomous effects may occur and highlight the muscle example (lines 229-231). Though outside the scope of the current study, we note that single-nuclei RNAseq of *eff-1* fusion mutants could address this interesting question.

The logic for the ELT-3 section seems a bit circular to me- ELT-3 targets are known to be enriched in hyp, hyp genes tend to be downregulated in *eff-1* mutants. Is there anything to suggest ELT-3 targets are more downregulated than other hyp-enriched genes? If so this would be useful to incorporate into Fig. S2.

We thank the reviewer for this helpful suggestion and have now included the average log₂ fold changes and average adjusted p-values for down DEGs associated or not associated with ELT-3 in Figure S4B. This analysis shows that ELT-3 associated downregulated genes exhibit more negative fold changes and more significant p-values than non-ELT-3 associated down DEGs.

Reviewer 2: This manuscript approaches an interesting and poorly understood question regarding the role of cell fusion in gene regulation and differentiation. The authors take advantage of a large-scale, developmentally programmed set of cell fusions during *C. elegans* embryogenesis in which 23 embryonic cells fuse to create the initial larval hyp7 epidermal syncytium. They generated a predicted null allele in the fusogen gene, *eff-1*, and compared it to two classic alleles, which are likely partial reduction-of-function alleles. Their null allele displayed comparable larval developmental delay to the two hypomorphs and the authors found a common set of misregulated genes in the *eff-1* mutants through RNA-seq. The cell fusion defective *eff-1* mutants have a gene expression pattern consistent with a developmental delay and failure to establish a larval transcriptome. Overall, this is an interesting and rigorous study and suitable for Development once the following issues are addressed.

We thank the reviewer for their supportive comments.

Major points

1. Lines 103-104. The authors indicate that the cell fusion defects in their *eff-1* null mutant persist to adulthood, but only L1 images are shown. Given that the hypomorphs were reported to have partial cell fusion during larval development, it would be valuable to include data showing the difference between adult null and hypomorph cell fusion.

We now include the relevant images as a new supplementary figure, Fig S1 and briefly discuss the effects of *eff-1* mutations on adult cell fusion (lines 116-120).

2. Given the developmental delay of the *eff-1* mutants and the large-scale oscillatory gene expression in the epidermis how did the authors ensure that the RNA samples were harvested at comparable developmental timepoints? This information is critical for interpreting the RNA-seq and smiFISH data.

As mentioned above we have updated our methods section to include this information. Because *eff-1* mutations inherently lead to developmental differences, we chose to sequence larva arrested in L1 diapause through starvation, a stalled developmental state in which all animals from all mutant backgrounds became synchronized without proceeding further into larval development. While there are potential caveats to this approach, we decided that synchronization was the best and most direct option for the transcriptomic comparison of fused vs. unfused worms. To clarify this approach, we have expanded the results and methods sections to detail how synchronization was accomplished prior to RNA collection (lines 140-143 and 322-325).

3. There are a number of translational fusions available for pre-cuticle components and various cuticle components, many of which are dysregulated in *eff-1* mutants. It would substantially strengthen the paper to look at localization of pre-cuticle components such as *noah-1* and *noah-2*, embryonic cuticle components like *dpy-14* or *dpy-17* (should be in L1 cuticle but not expressed in L1), and markers for other structures like furrows and annuli. Those experiments would demonstrate how this embryonic-like gene expression program affects cuticle structure.

We agree and thank the reviewer for this excellent suggestion, which addresses a key area for our planned future research. Investigating the localization of pre-cuticle and cuticle components is highly relevant and would certainly provide valuable mechanistic insights.

We note that we did use available CRISPR-generated translational fluorescent fusions for the pre-cuticle components *NOAH-1::mCherry* and *NOAH-2::GFP* in our *eff-1* mutant background. Unexpectedly, however, quantitative confocal imaging of these strains revealed no significant changes in overall protein levels or in the localization of these markers in L1 larvae compared to wild-type controls. This was surprising, especially considering the dramatic dysregulation of their corresponding transcripts in our RNA-seq data. The discrepancy between the transcriptional and protein-level data may suggest a complex post-transcriptional and/or post-translational regulatory mechanism, possibly involving the sequestration of excess mRNAs and/or increased protein turnover. Other explanations exist and carefully considered follow-up experiments to address this issue warrant a separate, dedicated study.

We also agree that creating and analyzing *DPY-14* and *DPY-17* strains are compelling future experiments. However, due to the focused scope of this manuscript as a research report, our primary goal was to present a comprehensive transcriptomic study of the global gene

expression changes that occur in the absence of cell fusion. The suggested experiments, while highly valuable, unfortunately fall outside the scope of this manuscript and would require significant time and resources for strain generation and detailed phenotyping.

Minor points

1. In hindsight it would have been better to generate clean *eff-1* P183L and S441L mutations by CRISPR in an N2 background to remove the confounding variable of other background mutations. That work is beyond the scope of this paper as the work is rigorous, but it would have allowed insight into gene regulation differences produced by the null vs the hypomorphic alleles. We agree as this would have allowed us to more directly explore differences between the point mutants and our null. This could be the basis of future work to investigate structural and functional domains within the *eff-1* protein, but we agree it is outside the scope of the current manuscript.
2. I may have missed it, but I didn't see any description/discussion of the Time to Hatch data in Fig 1E. It's worth noting that embryonic development seems to proceed normally, but larval development is delayed. As mentioned above, we now include a discussion of these data and the effects on embryonic vs. larval development (lines 122-133).
3. The species and gene names in the references need italicizing. We have corrected the references.

Second decision letter

MS ID#: dev.205089R1

MS TITLE: Epidermal cell fusion promotes the transition from an embryonic to a larval transcriptome in *C. elegans*

AUTHORS: David S. Fay, Owen H. Funk and Daniel L. Levy

Dear Dr Fay,

I am happy to tell you that your manuscript has been accepted for publication in *Development*, pending our standard publication integrity checks.

Reviewer 1

Advance summary and potential significance to field

After reading the revised version of this manuscript, I remain enthusiastic about the paper overall. The responses to my comments and those of the other reviewer mostly were textual and limited in scope such that some of the questions raised in both reviews persist. However, in my view the observations included still warrant publication and the disclaimers in the text and authors' arguments that resolving these issues would be better suited for future studies are reasonable.

Reviewer 2

Advance summary and potential significance to field

The authors have done a nice job addressing the reviewer critiques. I largely agree with their rebuttal. The one point that I feel should be added is their NOAH-1/NOAH-2 translational fusion

data. It seems an important point that the changes in pre-cuticle gene expression are not necessarily manifested at the protein level and they can set that up as a future direction.

Minor points:

1. There are several *C. elegans* or *Caenorhabditis elegans* or gene names that need italicizing in the references. ie. Lines 479, 500, 533, 576-577, 593.